# Instruction Decomposition and Action Alignment for Vision-and-Language Navigation

**Zihao Xin** [1]  **Wentong Li** [1 ‡]  **Yixuan Jiang** [1]  **Bin Wang** [2]  **Piji Li** [1]  **Jianke Zhu** [3]  **Jie Qin** [1]  **Shengjun Huang** [1 ‡]

## Abstract

Vision-and-Language Navigation (VLN) empowered by Multimodal Large Language Models (MLLMs) has shown great promise, yet remains challenging in long-horizon scenarios involving complex user instructions. Existing methods that continuously condition on full instructions incur substantial latency and suffer from instruction interference, where irrelevant textual cues can induce hallucinated actions. To address these limitations, we propose **IDEAL-VLN** (**I**nstruction **DE**composition and **A**ction a**L**ignment), a novel paradigm that reformulates VLN as a causal inference chain with two sequential steps: Semantic Anchoring and Action Alignment. We adopt a *Think-Before-Act* mechanism that first infers the immediate semantic anchor from the global context and then generates actions conditioned solely on this anchor. This design constructs an explicit information bottleneck that suppresses spurious correlations from irrelevant instruction. Moreover, we introduce a hierarchical correction mechanism that combines semantic-level thought correction with spatially-aware adaptive intervention. This design adjusts expert intervention probability based on geodesic distance, effectively defining a semantic safety boundary. To support this paradigm, we also construct an Instruction-Aligned Navigation Dataset with 160K image-text pairs. Extensive experiments show that IDEAL-VLN achieves state-of-the-art performance and strong robustness across major VLN benchmarks while significantly reducing inference costs. Code will be released at https://allenxinn.github.io/IDEALVLN.

[1]Nanjing University of Aeronautics and Astronautics, Nanjing, Jiangsu, China [2]Shandong University, Jinan, Shandong, China [3]Zhejiang University, Hangzhou, Zhejiang, China. Correspondence to: Wentong Li <wentong_li@nuaa.edu.cn>, Shengjun Huang <huangsj@nuaa.edu.cn>.

*Proceedings of the 43$^{rd}$ International Conference on Machine Learning*, Seoul, South Korea. PMLR 306, 2026. Copyright 2026 by the author(s).

## 1. Introduction

Vision-and-Language Navigation (VLN) (Wei et al., 2025b;a; Zhang et al., 2025a; Xin et al., 2026a) requires an embodied agent to interpret and execute natural language instructions within complex environments, relying exclusively on egocentric visual observations. With the rapid progress of Multimodal Large Language Models (MLLMs) (Liu et al., 2024; Wang et al., 2024a; Bai et al., 2025a), a growing line of work aims to leverage their robust generalization ability for VLN. Most contemporary paradigms focus on preserving long-horizon navigation consistency by strengthening memory modules (Wei et al., 2025a; Zhang et al., 2025a) or extending context window (Wei et al., 2025b; Zhang et al., 2025b). However, these approaches are inherently constrained by the prohibitive computational costs associated with visual tokens (Li et al., 2025a). Under restricted GPU memory budgets, aggressive compression of historical information inevitably results in the attrition of critical navigation cues. Critically, continuously conditioning the policy on the full, complex instruction introduces substantial semantic noise irrelevant to the current timestep. This instruction interference exacerbates model hallucinations, causing agents to yield spurious responses.

To address these challenges, we propose **IDEAL-VLN** (**I**nstruction **DE**composition and **AL**ignment), a new navigation paradigm that explicitly decouples understanding from execution. In contrast to prior end-to-end black-box mapping policies, we reformulate the VLN task as a *causal inference chain* with two sequential reasoning steps: ***Semantic Anchoring*** and ***Action Alignment***. To validate this factorization, we construct a large-scale Instruction-Aligned Navigation Dataset. Empirical analysis on this dataset reveals that existing VLN approaches generally lack the capacity for explicit comprehension of complex textual instructions, thereby corroborating the necessity of decoupling understanding from execution.

Based on the IDEAL-VLN paradigm, we introduce a *Think-Before-Act* mechanism, which departs from the conventional paradigm of passively receiving instructions and instead enables the agent to proactively reason about its immediate navigation goal before taking actions. At each decision point, the agent first *thinks* by generating a sub-instruction

that is strongly aligned with the current observation; this sub-instruction then serves as the sole short-term navigation objective for action prediction. This process effectively reduces the dimensionality of complex long-horizon navigation into a series of simple atomic sub-tasks, fundamentally blocking cross-temporal textual interference and improving decision-making reliability. To avoid unnecessary reasoning overhead, we further present an adaptive triggering strategy that uses action entropy to estimate the model's internal uncertainty and identify critical moments for re-planning.

However, training an agent with explicit planning capabilities introduces unique challenges, making existing imitation learning algorithms such as DAgger (Ross et al., 2011) inapplicable. First, conventional DAgger performs correction only at the action-level, whereas IDEAL-VLN critically depends on the intermediate thought process. In our paradigm, the sub-instruction acts as a semantically grounded latent variable in the causal chain. Second, existing intervention strategies typically rely on static or linearly decaying expert probabilities, failing to adapt to the non-linear impact of spatial location on semantic understanding in VLN tasks. To address these limitations, we propose a hierarchical correction mechanism that combines semantic-level thought correction with spatially-aware adaptive intervention. This mechanism jointly optimizes the causal inference chain from both cognitive and behavioral perspectives. It mitigates cascading errors caused by incorrect intermediate thoughts while balancing in-distribution exploration and out-of-distribution safety, thereby substantially enhancing robustness in complex unseen environments.

In summary, our main contributions are as follows:

- We propose IDEAL-VLN, a new VLN paradigm that formulates VLN as a causal inference chain consisting of *Semantic Anchoring* and *Action Alignment*, mitigating instruction interference in long-horizon navigation.

- We construct a large-scale Instruction-Aligned Navigation dataset and introduce a Think-Before-Act mechanism with a zero-overhead adaptive triggering strategy to suppress navigation hallucinations.

- We present a hierarchical correction mechanism that combines semantic-level thought correction with spatially-aware adaptive intervention, jointly improving thought reliability and action stability in unseen environments.

## 2. Related Work

### 2.1. Multimodal Large Language Models

Early multimodal exploration primarily focus on projecting visual representations into the semantic space of pre-trained large language models (LLMs). BLIP-2 (Li et al., 2023) stands as a pioneering work, introducing the Q-Former to enable efficient cross-modal alignment with minimal supervision. Building on this line, the LLaVA series (Liu et al., 2023b;a; 2024; Li et al., 2024) systematically establish the visual instruction tuning paradigm, and subsequent works (Li et al., 2025b; Shi et al., 2024; Yuan et al., 2024; Chen et al., 2024b; Tong et al., 2024; Li et al., 2025a) further advance model capacity and training recipes. In parallel, Qwen-VL family (Wang et al., 2024a; Bai et al., 2025b) address the limitations of input resolution and temporal modeling by introducing a dynamic resolution mechanism. More recently, Qwen3-VL (Bai et al., 2025a) proposes interleaved MROPE, which balances positional frequency distributions across temporal, horizontal, and vertical dimensions and helps mitigate positional bias in long-video modeling. Despite this progress, recent studies (Gholami et al., 2025; Qi et al., 2025a; Yuan et al., 2025) indicate that current MLLMs still struggle to establish robust geometric correspondence between egocentric partial observations and the environment, an ability that is crucial for embodied navigation.

### 2.2. Vision-and-Language Navigation

Vision-and-Language Navigation (VLN) requires agents to reach target locations by following natural language instructions from egocentric visual observations. Early VLN methods mainly adopt Seq2Seq architectures (Guhur et al., 2021; Irshad et al., 2022; Chen et al., 2023), while recent approaches leverage Large Language Models for high-level planning, such as NavGPT (Zhou et al., 2024) and InstructNav (Long et al., 2024). NaVid (Zhang et al., 2024a) pioneer the paradigm of VLM-based end-to-end action prediction but suffer from severe challenges regarding context length limitation. To mitigate this, StreamVLN (Wei et al., 2025b) uses a slow-fast memory mechanism with sliding window modeling with voxel compression. Meanwhile, InternVLA-N1 (Wei et al., 2025a) and OmniNav (Xue et al., 2025) adopt dual-system architectures, decoupling low-frequency high-level planning from high-frequency low-level execution. Nevertheless, most VLN models (Zhang et al., 2024a; Qi et al., 2025b; Wei et al., 2025b; Cheng et al., 2025; Xin et al., 2026b) have yet to fundamentally resolve the computational efficiency issues associated with long-term memory. Recently, AgentVLN (Xin et al., 2026a) presents a "VLM-as-Brain" framework that connects 2D visual reasoning with 3D planning through modular skills, enabling more robust and efficient long-horizon VLN.

## 3. Method

### 3.1. Divide and Conquer

In unseen environments, VLN requires an embodied agent to navigate to target locations following complex natural

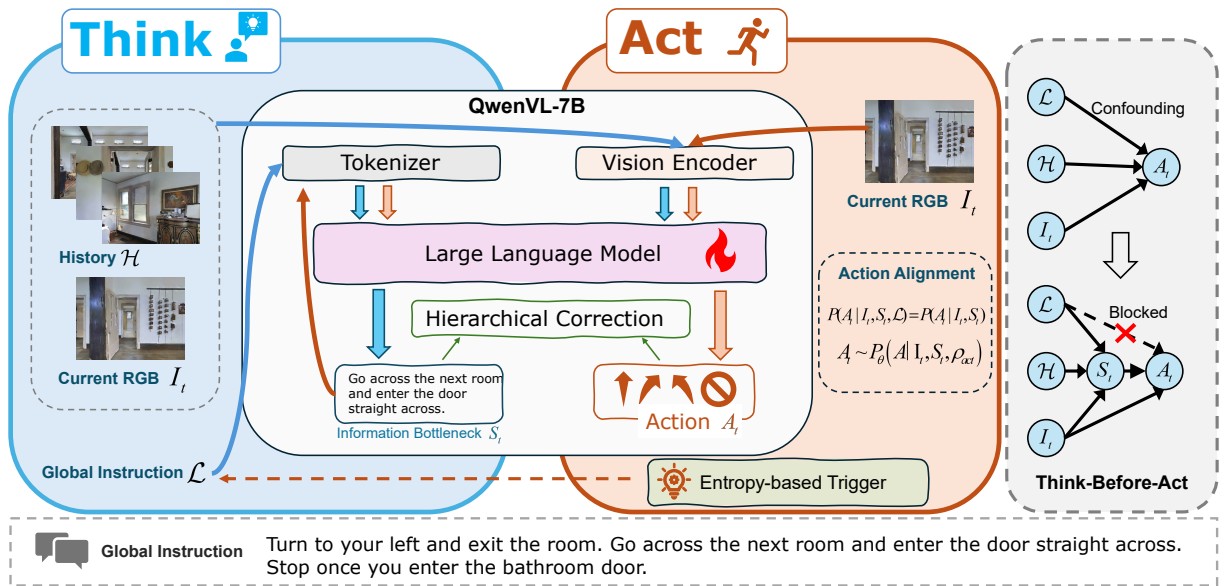

*Figure 1.* Overview of our IDEAL-VLN. We present a novel *Think-Before-Act* paradigm that formulates VLN as a causal inference chain, comprising two sequential steps: Semantic Anchoring (*Think*, blue) and Action Alignment (*Act*, orange). The model first generates a concise semantic anchor (sub-instruction) $S_t$ from the global instruction $\mathcal{L}$ and history $\mathcal{H}$, triggered only at critical points by an entropy-based signal. Actions $A_t$ are then predicted from $(I_t, S_t)$, while a causal intervention blocks the direct path $\mathcal{L} \rightarrow A_t$, enforcing $A_t \perp \mathcal{L} \mid (S_t, I_t)$ to reduce instruction interference. A hierarchical correction module jointly corrects both $S_t$ and $A_t$ during training.

language instructions, relying solely on egocentric observations. However, at any specific time step $t$, the agent's current action correlates only with a *local* portion of the instruction rather than the entire text. Nevertheless, most existing VLN frameworks adopt an end-to-end full-instruction conditioning paradigm: at time step $t$, the policy takes the historical observation sequence $O_{1:t} = \{I_1, ..., I_t\}$ together with the complete instruction $\mathcal{L}$ and directly predicts an action distribution,

$$A_t = \pi_\theta(O_{1:t}, \mathcal{L}). \quad (1)$$

Conditioning on the full long-form instruction $\mathcal{L}$ at every step can introduce substantial instruction interference, where non-current instruction segments act as semantic noise and amplify hallucinations, ultimately degrading action reliability. To mitigate the instruction interference, we propose **IDEAL-VLN**, which models navigation as a *causal inference chain*, abstracting the navigation decision process into a causal graph comprising four nodes: the global instruction $\mathcal{L}$, historical visual observations $O_{1:t}$, the current intent $S_t$, and the action $A_t$. We formalize this causal chain via two steps: *Semantic Anchoring* and *Action Alignment*.

**Semantic Anchoring.** This step maps the global context to a local intent by explicitly extracting the mediator $S_t$. Given the full observation history $O_{1:t}$ and instruction $\mathcal{L}$, the model first infers the current navigation state and predict a sub-instruction $S_t$ that is most relevant for the current decision. We introduce a decomposition prompt $\rho_{decomp}$ to facilitate this process, formulated as sampling a semantic

anchor from the conditional probability distribution:

$$S_t \sim P_\theta\left(S \mid O_{1:t}, \mathcal{L}, \rho_{decomp}\right). \quad (2)$$

In this phase, $S_t$ serves as an information bottleneck; it is optimized to preserve decision-critical semantics while discarding redundant content in $\mathcal{L}$ that refers to past steps or future goals.

**Action Alignment.** In this subsequent step, we enforce that action generation depends on the global instruction *only through* the mediator $S_t$. To this end, we perform a causal intervention during the action generation. The action generation process is modeled as the result of performing the do-operator on the mediator $S_t$:

$$A_t \sim P_\theta\left(A \mid I_t, do(S_t), \rho_{act}\right), \quad (3)$$

where $\rho_{act}$ is an execution prompt that conditions the model solely on the current observation $I_t$ and the semantic anchor $S_t$. The intervention $do(S_t)$ blocks the direct pathway $\mathcal{L} \rightarrow A_t$, implying that during execution the action is conditionally independent of the global instruction given $(I_t, S_t)$, *i.e.*, $P(A_t|I_t, S_t, \mathcal{L}) = P(A_t|I_t, S_t)$. As a result, any influence of $\mathcal{L}$ on $A_t$ must be transmitted through $S_t$. This structured constraint makes the model's intent explicit, reduces cross-temporal textual interference, and improves the robustness of action generation in long-horizon navigation.

Since existing VLN datasets provide only trajectory-level instruction annotations and lack frame-level correspondence,

we construct a large-scale **Instruction-Aligned Navigation Dataset**. The dataset is built on the Matterport3D (MP3D) (Chang et al., 2017) simulation environment, covering 61 diverse indoor scenes. To ensure instruction realism and complexity, we inherit the original full navigation instructions $\mathcal{L}$ and waypoints from R2R (Anderson et al., 2018) and RxR (Ku et al., 2020). We then use the Habitat simulator to re-collect egocentric visual images along the trajectories, yielding 160K image–sub-instruction pairs. ***The detailed data construction pipeline is described in Appendix C.***

## 3.2. Think-Before-Act

We instantiate the causal chain of navigation into two steps: Semantic Anchoring and Action Alignment. This *Think-Before-Act* mechanism decouples a long-horizon instruction-following navigation task into a sequence of concise sub-tasks. During navigation, instead of passively conditioning on the full instruction, the agent proactively reasons about the current state and generates a sub-instruction $S_t$, which serves as the simple short-term goal for action alignment. Therefore, this scheme avoids repeatedly processing redundant historical frames: the policy conditions only on the current observation $I_t$ and the compact semantic anchor $S_t$. This design improves inference efficiency and, by suppressing instruction segments irrelevant to the current context, reduces hallucinations caused by cross-temporal textual interference. Once the agent determines that the current sub-task has been completed, it triggers another round of reasoning to produce the next sub-instruction $S_{t+1}$, enabling a smooth transition of navigation goals.

The core challenge of the thinking mechanism lies in *when to trigger*. Excessively frequent reasoning increases latency, while overly sparse updates may lead to navigation failure. To ensure accuracy while achieving zero inference overhead, we propose mining the implicit uncertainty inherent in the model's own action outputs. We define the navigation action space as $\mathcal{A} = \{\texttt{Forward}, \texttt{Turn\_Left}, \texttt{Turn\_Right}, \texttt{Stop}\}$, with rotation actions fixed at a step size of $15°$. At each time step $t$, the model predicts a Micro-action Sequence of $N$ actions, denoted as $\mathbf{a}_t = \{a_t^1, \ldots, a_t^N\}$. By analyzing the distribution patterns of $\mathbf{a}_t$, we design two triggering conditions.

**Confusion-Induced Dithering.** When the model lacks a clear understanding of the current instruction or becomes disoriented, it often exhibits tentative oscillating behavior. If the action sequence $\mathbf{a}_t$ contains both left and right turn instructions simultaneously:

$$\exists i, j \in [1, N], \text{ s.t. } a_t^i = \texttt{Left} \wedge a_t^j = \texttt{Right}. \quad (4)$$

This indicates that the model is experiencing a decision conflict, characterized by oscillating attention. In this scenario,

the system detects this state of uncertainty and proactively triggers the reasoning mechanism to recalibrate the sub-instruction based on global information.

**Scene Transition.** A significant rotation by the agent within a short duration implies imminent entry into a new observational scene. If the action sequence $\mathbf{a}_t$ contains $k$ or more rotation actions in the same direction. At this point, the semantic content of the egocentric image $I_t$ undergoes a significant shift. To prevent misalignment between the outdated sub-instruction and the new scene, the system interprets this as a scene-switching signal and triggers the reasoning mechanism to generate a navigation goal adapted to the new field of view. Through this adaptive mechanism, IDEAL-VLN identifies critical decision points during navigation without introducing additional models or computational overhead, thereby achieving simultaneous enhancements in both computational efficiency and navigation accuracy.

## 3.3. Hierarchical Correction Mechanism

Unlike traditional imitation learning, which focuses solely on action-level correction, our IDEAL-VLN paradigm necessitates the joint error correction of both the model's thinking process and its action generation. To this end, we propose Hierarchical Correction Mechanism(HCM), comprising two components: Semantic-Level Thought Correction(STC) and Spatially-aware Adaptive Intervention(SAI).

**Semantic-Level Thought Correction.** During error-correction data collection, it is crucial to assess whether the predicted sub-instruction $S_{\text{pred}}$ is semantically consistent with the expert intent at timestep $t$. Given the agent's physical location $p_t$, we query the instruction-aligned navigation dataset to find the two nearest waypoints on the reference trajectory and extract the corresponding instruction segments. We denote this expert sub-instruction as $S_{\text{gt}}$. Let $W(S)$ denote the set of valid words in instruction $S$, the correctness score of the prediction is measured by the word-overlap metric $\mathcal{R}$:

$$\mathcal{R}(S_{pred}, S_{gt}) = \frac{|W(S_{pred}) \cap W(S_{gt})|}{|W(S_{pred})|}. \quad (5)$$

Given a threshold $\tau$, if $\mathcal{R} \geq \tau$, we consider the model to be "thinking correctly" and keep $S_{pred}$ for subsequent action generation. Otherwise, the model is deemed to exhibit cognitive hallucination. In this case, we perform Thought Forcing: the erroneous $S_{pred}$ is replaced with the expert label $S_{gt}$ as input to the action module, preventing error propagation along the causal chain. Meanwhile, we add this negative sample pair $\{(O_{1:t}, \mathcal{L}) \rightarrow S_{gt}\}$ to the error correction dataset $\mathcal{D}_{error}$.

**Spatially-Aware Adaptive Intervention.** Existing error correction strategies typically employ fixed expert probabilities. However, in complex VLN, the agent's tolerance

for deviation from the reference trajectory is highly state-dependent. Under the *Think-Before-Act* paradigm, balancing in-distribution exploration and out-of-distribution correction becomes a critical trade-off. In IDEAL-VLN, action reliability strictly depends on the upstream instruction decomposition module $P(S_t|O_{1:t}, \mathcal{L})$. When the agent's position $p_t$ deviates substantially from the reference path, the current egocentric observation $I_t$ becomes completely detached from the semantics described by the global instruction $\mathcal{L}$. This visual-text semantic disconnection causes the instruction decomposition module to fail in extracting valid sub-instructions from the irrelevant visual context. Such upstream cognitive errors propagate though the causal chain to the downstream action module, leading to unrecoverable navigation failure. Conversely, adopting an overly aggressive intervention strategy hampers the model's ability to recover from perturbations.

Motivated by this analysis, we propose a spatially-aware adaptive intervention strategy. Specially, we model the expert intervention probability $P_{int}$ as a non-linear function of the cross-track error $d_t$, using a soft gating rule:

$$P_{int}(d_t) = \frac{1}{1 + e^{-k \cdot (d_t - \mu)}}, \qquad (6)$$

where $d_t$ is the geodesic distance from the agent's current position to the nearest waypoint on the reference trajectory, $k$ controls sensitivity, and $\mu$ defines the radius of the semantic safety boundary. When $d_t < \mu$, the model is considered to be in a recoverable region, and $P_{int}$ approaches 0, allowing autonomous exploration based on $S_{pred}$, $A_{pred}$ and enabling the policy to learn recovery behaviors from small deviations. Overall, SAI ensures that instruction decomposition operates under semantically valid support when necessary, while preserving sufficient exploration to improve robustness to real-world physical noise.

**Difference from CoT.** Unlike Chain-of-Thought (CoT), which typically relies on sequential supervision over static datasets to generate free-form rationales, HCM provides online, closed-loop correction for embodied decision-making. Its supervision is grounded in structured task variables: STC corrects the latent sub-instruction $S_t$ to preserve semantic validity, while SAI adjusts expert intervention according to geodesic deviation to keep the agent within a valid data distribution. In this way, HCM jointly corrects thoughts and actions during policy execution, preventing error propagation along the causal chain in dynamic physical environments.

# 4. Experiment

## 4.1. Experimental Setup

As in prior research (Zhang et al., 2025b; Cheng et al., 2025), we conduct experiments on the R2R (Anderson et al., 2018) and RxR (Ku et al., 2020) benchmark datasets. We utilize the Habitat simulator to capture egocentric visual observations within the Matterport3D (Chang et al., 2017) environment. Furthermore, image augmentation strategies are incorporated during the training phase to bolster the model's generalization. To optimize the model's instruction-following proficiency and mitigate catastrophic forgetting during fine-tuning, we incorporate two auxiliary datasets during the error-correction fine-tuning phase: ScaleVLN-150K (Wang et al., 2023b) and LLaVA-Video-178K (Zhang et al., 2024b). Specifically, we randomly select 80k video QA samples from LLaVA-Video-178K to ensure that, while mastering navigation skills, the model retains its capacity for recognition and reasoning regarding general visual objects. Simultaneously, ScaleVLN-150K is utilized to supplement diverse navigation instructions, enhancing the model's adaptability to linguistic styles. We report five standard metrics: Success Rate (SR), Success weighted by Path Length (SPL), Navigation Error (NE), Oracle Success (OS), and Normalized Dynamic Time Warping (nDTW).

## 4.2. Implementation Details

Our model is built upon the Qwen2.5-VL-7B (Bai et al., 2025b) model. We employ the AdamW optimizer for training, with a global batch size set to 128. We adopt a cosine annealing schedule for learning rate adjustment, initializing the peak learning rate at $2 \times 10^{-5}$ and setting a warmup ratio of 0.03 to ensure stability during the early training stages. Additionally, the model's visual encoder and multimodal projection layer are frozen. Regarding the hierarchical correction mechanism, we set the correction threshold $\tau = 0.8$, the semantic safety radius $\mu = 1.5$ meters, and the sensitivity coefficient $k = 3$. This configuration ensures a rapid response in intervention probability as the agent approaches the safety boundary. All experiments were conducted on a cluster of 8 NVIDIA H100 GPUs. For more implementation details, please refer to Appendix C.

## 4.3. Main Results

We first compare IDEAL-VLN with state-of-the-art (SOTA) methods on the R2R-CE and RxR-CE Val-Unseen benchmarks. The baselines include both waypoint-prediction methods with multi-sensor inputs and large-scale visual navigation models, with detailed results reported in Table 1 and Table 2. Benefiting from the Think-Before-Act decomposition paradigm, IDEAL-VLN achieves new state-of-the-art performance on both benchmarks. Specifically, on R2R-CE Val-Unseen, IDEAL-VLN reaches an SR of 59.8%, substantially outperforming existing monocular navigation methods. More importantly, on the more challenging RxR-CE Val-Unseen benchmark, which features longer instructions and more complex trajectories, our method achieves an SR of 59.1%, demonstrating strong robustness in long-horizon navigation. These results show that decomposing long in-

*Table 1.* Comparison with SOTA methods on the R2R-CE Val-Unseen benchmark. † indicates models trained with additional large-scale datasets. IDEAL-VLN achieves the best overall performance among monocular RGB-based methods, even surpassing several approaches that rely on extra training data, global priors, or multi-sensor inputs.

| Method | RGB | Pano. | Depth | Odo. | NE ↓ | OS ↑ | SR ↑ | SPL ↑ |
|---|---|---|---|---|---|---|---|---|
| HPN+DN (Krantz et al., 2021) | | ✓ | ✓ | ✓ | 6.31 | 40.0 | 36.0 | 34.0 |
| VLN BERT (Hong et al., 2022) | | ✓ | ✓ | ✓ | 5.74 | 53.0 | 44.0 | 39.0 |
| CMA (Hong et al., 2022) | | ✓ | ✓ | ✓ | 6.20 | 52.0 | 41.0 | 36.0 |
| Reborn (An et al., 2022) | | ✓ | ✓ | ✓ | 5.40 | 57.0 | 50.0 | 46.0 |
| Ego$^2$-Map (Hong et al., 2023) | | ✓ | ✓ | ✓ | 5.54 | 56.0 | 47.0 | 41.0 |
| DreamWalker (Wang et al., 2023a) | | ✓ | ✓ | ✓ | 5.53 | 59.0 | 49.0 | 44.0 |
| HAMT+ScaleVLN (Wang et al., 2023b) | | ✓ | ✓ | ✓ | 4.80 | - | 55.0 | 51.0 |
| ETPNav (An et al., 2023) | | ✓ | ✓ | ✓ | 4.71 | 65.0 | 57.0 | 49.0 |
| Seq2Seq (Krantz et al., 2020) | ✓ | | | ✓ | 7.77 | 37.0 | 25.0 | 22.0 |
| RGB-CMA (Krantz et al., 2020) | ✓ | | | | 9.55 | 10.0 | 5.0 | 4.0 |
| AG-CMTP (Chen et al., 2021) | | ✓ | ✓ | ✓ | 7.90 | 39.0 | 23.0 | 19.0 |
| R2R-CMTP (Chen et al., 2021) | | ✓ | ✓ | ✓ | 7.90 | 38.0 | 26.0 | 22.0 |
| LAW (Raychaudhuri et al., 2021) | ✓ | | ✓ | ✓ | 6.83 | 44.0 | 35.0 | 31.0 |
| CM2 (Georgakis et al., 2022) | ✓ | | ✓ | ✓ | 7.02 | 41.0 | 34.0 | 27.0 |
| WS-MGMap (Chen et al., 2022) | ✓ | | ✓ | ✓ | 6.28 | 47.0 | 38.0 | 34.0 |
| ETPNav+FF (Wang et al., 2024b) | ✓ | | ✓ | ✓ | 5.95 | 55.8 | 44.9 | 30.4 |
| AO-Planner (Chen et al., 2024a) | | ✓ | ✓ | | 5.55 | 59.0 | 47.0 | 33.0 |
| NaVid (Zhang et al., 2024a) | ✓ | | | | 5.47 | 49.0 | 37.0 | 35.0 |
| VLN-R1 (Qi et al., 2025b) | ✓ | | | | 7.90 | 41.2 | 30.2 | 21.8 |
| StreamVLN (Wei et al., 2025b) | ✓ | | | | 6.05 | 53.8 | 45.5 | 41.6 |
| **IDEAL-VLN** | ✓ | | | | **5.05** | **62.3** | **56.7** | **51.6** |
| Uni-NaVid† (Zhang et al., 2025b) | ✓ | | | | 5.58 | 53.3 | 47.0 | 42.7 |
| NaVILA† (Cheng et al., 2025) | ✓ | | | | 5.22 | 62.5 | 54.0 | 49.0 |
| StreamVLN† (Wei et al., 2025b) | ✓ | | | | 5.10 | 64.0 | 55.7 | 50.9 |
| InternVLA-N1† (Contributors, 2025) | ✓ | | | | 4.89 | 60.6 | 55.4 | 52.1 |
| NavFoM† (Zhang et al., 2025a) | ✓ | | | | 5.01 | 64.9 | 56.2 | 51.2 |
| **IDEAL-VLN†** | ✓ | | | | **4.62** | **64.9** | **59.8** | **52.9** |

structions into atomic sub-instructions effectively reduces cumulative errors in long-sequence reasoning, thereby improving both SR and SPL. Notably, even when trained exclusively on R2R-CE and RxR-CE, IDEAL-VLN still achieves 56.7% and 55.9% SR, respectively, surpassing strong baselines such as StreamVLN (Wei et al., 2025b), which rely on additional datasets such as ScaleVLN and extensive multimodal data.

### 4.4. Ablation Study

**Component-wise Effects.** To evaluate the contribution of each core component in IDEAL-VLN, we conduct incremental ablation studies on the challenging RxR-CE Val-Unseen benchmark, using a Qwen2.5-VL-7B-based end-to-end full-instruction model as the baseline. As shown in Figure 2, introducing the Think-Before-Act (TBA) mechanism improves SR by 8.4% and significantly enhances path quality, confirming that decomposing navigation into semantic anchoring and action alignment effectively reduces interference from non-current instruction segments. Build-

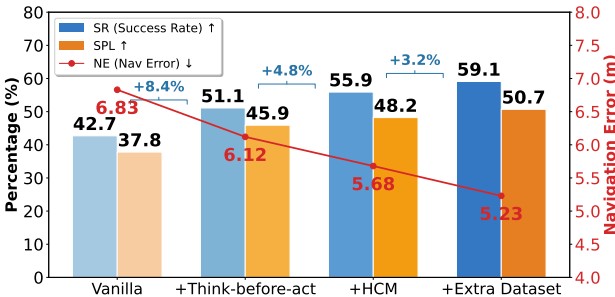

*Figure 2.* Ablation study on RxR-CE Val-Unseen. Each component of IDEAL-VLN progressively improves SR and SPL while reducing NE.

ing on TBA, the Hierarchical Correction Mechanism (HCM) brings an additional 4.8% SR improvement by correcting both reasoning hallucinations and trajectory deviations during exploration, thereby improving robustness in unseen environments. Finally, multimodal joint training further increases SR by 3.2%, indicating that preserving general multimodal understanding with LLaVA-Video and enhancing instruction diversity with ScaleVLN are both beneficial

*Table 2.* Comparison results with SOTA methods on the Val-Unseen dataset for RxR-CE.

| Method | RGB | Pano. | Depth | Odo. | NE ↓ | SR ↑ | SPL ↑ | nDTW ↑ |
|---|---|---|---|---|---|---|---|---|
| VLN BERT (Hong et al., 2022) | | ✓ | ✓ | ✓ | 8.98 | 27.0 | 22.6 | 46.7 |
| CMA (Hong et al., 2022) | | ✓ | ✓ | ✓ | 8.76 | 26.5 | 22.1 | 47.0 |
| Reborn (An et al., 2022) | | ✓ | ✓ | ✓ | 5.98 | 48.6 | 42.0 | 63.3 |
| ETPNav (An et al., 2023) | | ✓ | ✓ | ✓ | 5.64 | 54.7 | 44.8 | 61.9 |
| Seq2Seq (Krantz et al., 2020) | ✓ | | | ✓ | 12.10 | 13.9 | 11.9 | 30.8 |
| LAW (Raychaudhuri et al., 2021) | ✓ | | | ✓ | ✓ | 10.90 | 8.0 | 8.0 | 38.0 |
| ETPNav+FF (Wang et al., 2024b) | ✓ | | | ✓ | ✓ | 8.79 | 25.5 | 18.1 | - |
| AO-Planner (Chen et al., 2024a) | | ✓ | | ✓ | 7.06 | 43.3 | 30.5 | 50.1 |
| VLN-R1 (Qi et al., 2025b) | ✓ | | | | 9.1 | 22.7 | 17.6 | - |
| **IDEAL-VLN** | ✓ | | | | **5.68** | **55.9** | **48.2** | **63.7** |
| Uni-NaVid† (Zhang et al., 2025b) | ✓ | | | | 6.24 | 48.7 | 40.9 | - |
| NaVILA† (Cheng et al., 2025) | ✓ | | | | 6.77 | 49.3 | 44.0 | 58.8 |
| StreamVLN† (Wei et al., 2025b) | ✓ | | | | 6.16 | 51.8 | 45.0 | 61.9 |
| InternVLA-N1† (Contributors, 2025) | ✓ | | | | 6.41 | 49.5 | 41.8 | 62.6 |
| NavFoM† (Zhang et al., 2025a) | ✓ | | | | 5.51 | 57.4 | 49.4 | 60.2 |
| **IDEAL-VLN†** | ✓ | | | | **5.23** | **59.1** | **50.7** | **64.5** |

*Table 3.* Results of different triggering strategies for Think-Before-Act on the RxR benchmark.

| Method | Extra Mem. | NE ↓ | SR↑ | SPL↑ |
|---|---|---|---|---|
| Vanilla | - | 6.83 | 42.7 | 37.8 |
| LLM-based | - | 7.05 | 35.8 | 29.2 |
| Siglip-based | 0.9 G | 6.18 | 51.5 | 46.0 |
| Entropy-based | - | 6.12 | 51.1 | 45.9 |

*Table 4.* Results of different thought-triggering mechanisms.

| Method | NE ↓ | SR↑ | SPL↑ |
|---|---|---|---|
| Baseline | 6.12 | 51.1 | 45.9 |
| DAgger (Wei et al., 2025b) | 6.08 | 51.6 | 46.3 |
| STC | 5.79 | 53.2 | 47.1 |
| HCM | **5.68** | **55.9** | **48.2** |

*Table 5.* Comparison of similarity metric used in STC. WOR denotes the word-overlap ratio.

| Method | Extra Mem. | Extra FLOPs | SR |
|---|---|---|---|
| Sentence-BERT | ∼500MB | ∼10G | 56.1% |
| WOR ($\tau = 0.9$) | None | None | 55.2% |
| WOR ($\tau = 0.7$) | None | None | 55.8% |

Action-Entropy-based strategy achieves competitive performance with no additional computational cost.

**Effect of Hierarchical Correction Mechanism.** We evaluate the hierarchical correction mechanism in Table 4, where the baseline denotes the model without error correction. Compared with DAgger, which only corrects action-level errors, our Semantic-Level Thought Correction (STC) explicitly supervises the intermediate sub-instruction and improves SR by 1.6%. This shows that correcting the thought phase is crucial under the Think-Before-Act paradigm, as erroneous sub-instructions can otherwise contaminate the correction data and propagate errors along the causal chain. By further incorporating Spatially-Aware Adaptive Intervention (SAI), the full HCM achieves 55.9% SR and reduces NE to 5.68, yielding a 4.8% SR improvement over the baseline. These gains indicate that SAI effectively alleviates visual-text semantic misalignment by preventing the agent from drifting into regions where observations no longer support valid instruction decomposition.

**Similarity Metric in STC.** We compare the word-overlap ratio (WOR) against Sentence-BERT (Reimers & Gurevych, 2019) in the STC module of HCM, as shown in Table 5. Although WOR is less robust to Sentence-BERT, the threshold $\tau$ serves as a lightweight safety gate rather than a strict se-

for complex navigation.

**Effect of Think-Before-Act.** We evaluate the Think-Before-Act mechanism and its triggering strategies on the RxR benchmark. As shown in Table 3, we compare a vanilla full-instruction baseline with three variants: (1) LLM Confidence-based, which uses the LLM's output probabilities for triggering; (2) SigLIP-based, which employs SigLIP-so400m (Zhai et al., 2023) to compute the feature similarity between the current frame $I_t$ and the sub-instruction $S_t$; and (3) Action-Entropy-based, our adaptive triggering strategy. Think-Before-Act improves the baseline by 8.4% SR and 8.1% SPL, reaching 51.1% SR and 45.9% SPL, and also surpasses StreamVLN trained with EnvDrop and DAgger (48.6% SR). These results show that semantic re-anchoring reduces irrelevant instruction interference and navigation hallucinations. LLM Confidence-based triggering suffers from overconfident self-evaluation, while SigLIP-based triggering adds memory and latency overhead. In contrast, our

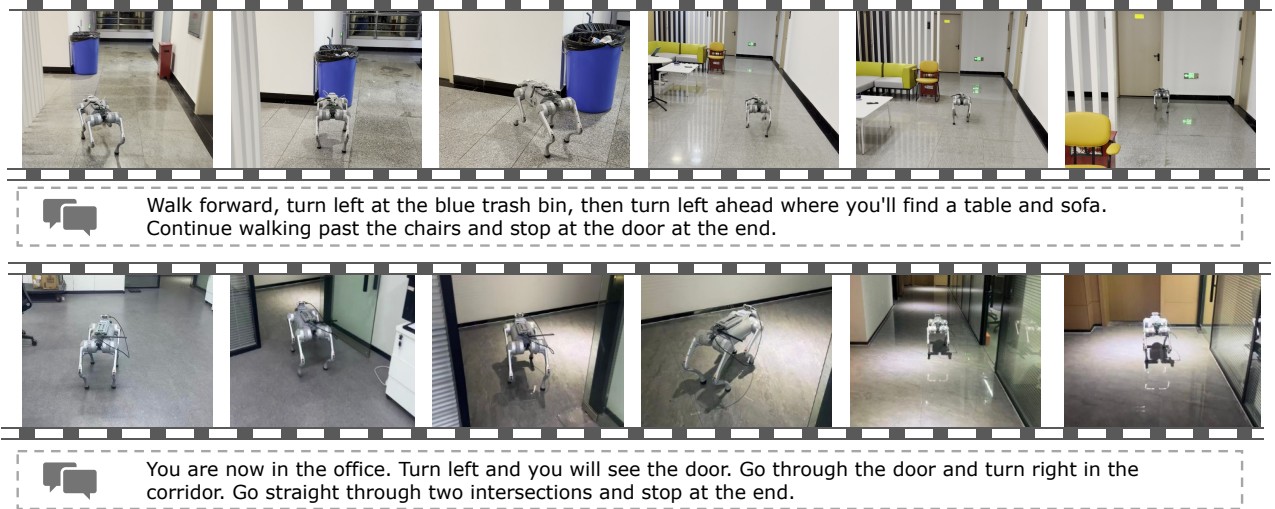

*Figure 3.* Qualitative results in real-world scenarios. IDEAL-VLN follows long-horizon instructions and generates feasible navigation behaviors on the Unitree Go2 in unseen office environments.

*Table 6.* Per-episode token and computational overhead comparison between full-instruction baseline and IDEAL-VLN.

| Stage | Tokens /Step | Steps | Total Tokens | Prefill FLOPs |
|---|---|---|---|---|
| Baseline (action) | 7K | 21 | 147K | ∼2,310T |
| IDEAL-VLN (think) | 7K | 5 | 35K | ∼550T |
| IDEAL-VLN (action) | 2K | 21 | 42K | ∼630T |

mantic metric. Since navigation instructions mainly consist of structured directional vocabulary and object landmarks, paraphrasing is relatively limited. Sentence-BERT brings marginal gains but introduces substantial computational overhead. We adopt WOR for its simplicity and efficiency.

### 4.5. Efficiency Analysis

IDEAL-VLN also significantly reduces token overhead compared with full-instruction paradigms. As shown in Table 6, conventional methods process the full instruction and historical context at every action step, consuming about 7K tokens per step and 147K tokens over 21 action chunks. In contrast, IDEAL-VLN decouples navigation into 5 low-frequency thinking steps and 21 lightweight action steps. Since action alignment only uses local observations and sub-instructions, the total token cost is reduced to 77K, nearly half of the conventional paradigm.

Furthermore, we effectively mask the thinking latency at the system implementation level as shwon in Figure 4. In real-world robotic deployments, the model outputs an action sequence at each decision step, which requires approximately 3 to 5 seconds for the physical agent to execute. Because the sub-task reasoning (which takes roughly 2 seconds) is performed concurrently while the agent is in motion, the

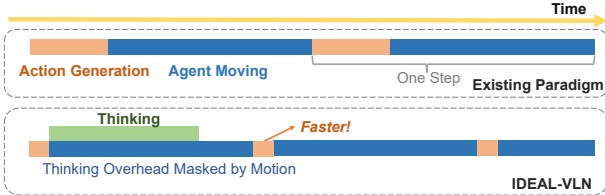

*Figure 4.* Illustration of latency masking in IDEAL-VLN. Sub-task reasoning is executed during agent motion, allowing the thinking overhead to be hidden by physical execution.

physical execution of the action sequence completely masks the thinking latency. Once the agent reaches the designated waypoint, the subsequent action sequence is generated based on the newly inferred sub-task. Given that the action generation overhead of IDEAL-VLN is merely one-third of that in existing paradigms, our method facilitates highly efficient real-time navigation.

### 4.6. Quantitative Analysis of Instruction Alignment

To evaluate the inherent navigation instruction analysis capabilities of MLLMs, we construct a validation split for the Instruction-Aligned Navigation Dataset from RxR-Val-Unseen using the same protocol. We evaluate Qwen2.5-VL (Bai et al., 2025b) and Qwen3-VL (Bai et al., 2025a) using the keyword overlap rate $\mathcal{R}(S_{pred}, S_{gt})$ as the evaluation metric. The results are shown in the Figure 5. The results show that general-purpose VLMs up to 72B parameters still struggle with fine-grained visual-text alignment in egocentric navigation, while stronger alignment emerges at the 235B scale. StreamVLN (Wei et al., 2025b) achieves only 29.31% accuracy, indicating limited understanding of the current navigation intent. In contrast, fine-

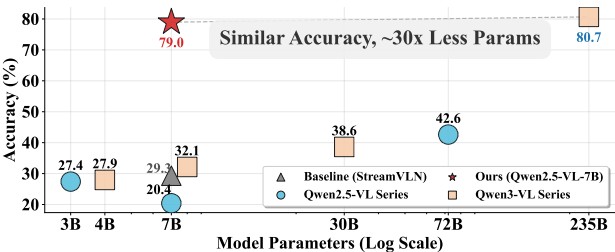

*Figure 5.* Instruction-alignment accuracy of different MLLMs. Our 7B model achieves performance comparable to the 235B Qwen3-VL model with about 30× fewer parameters.

tuning Qwen2.5-VL-7B on our dataset improves accuracy to 78.96%, achieving performance comparable to Qwen3-VL-235B with about 30× fewer parameters. We further find that around 15% of errors come from distant cue interference, where the model incorrectly associates faraway landmarks with the current sub-task; we mitigate this with a sequential coherence constraint during inference. For fair comparison with prior work (Wei et al., 2025b;a), we adopt Qwen2.5-VL-7B as our backbone.

### 4.7. Real-world Deployment

To validate the generalization potential of simulation-trained policies, we build a real-world navigation platform based on the Unitree Go2 quadruped robot. An ego-centric RealSense D455 camera is mounted at the front of the robot to capture RGB observations. Considering the high computational cost of large-model inference, we adopt an edge-cloud collaborative architecture: a Jetson Orin module on the robot runs a lightweight ASR model to transcribe human voice commands into text, and then sends the instruction together with the current ego-centric image $I_t$ to a server equipped with a single NVIDIA RTX 4090 GPU. The server hosts IDEAL-VLN and returns the predicted action sequence to the robot for execution, with the response loop completed within 0.2 seconds.

We evaluate IDEAL-VLN in unseen real-world office environments under a strict zero-shot sim-to-real setting, without using any real-world data for fine-tuning. Although inherent domain gaps exist between the Habitat simulator and the physical world in terms of visual rendering, such as lighting and textures, the results in Figure 3 confirm the adaptability of our approach. Even when facing high-frequency visual artifacts that are difficult to reproduce in simulation, such as specular reflections on polished floors, the agent accurately parses instructions and plans navigation paths under real-world physical constraints.

### 5. Conclusion

This paper presents IDEAL-VLN, a new VLN framework that mitigates instruction interference and reduces memory overhead in long-horizon navigation. We reformulate VLN as a causal inference chain with Semantic Anchoring and Action Alignment, and introduce a Think-Before-Act mechanism to decompose global instructions into local semantic anchors for action generation. To further improve efficiency and robustness, we propose an action-entropy-based trigger for adaptive re-anchoring and a hierarchical correction mechanism that jointly supervises thoughts and actions within a semantic safety boundary. We also construct an Instruction-Aligned Navigation Dataset for fine-grained instruction-perception alignment. Experiments show that IDEAL-VLN achieves state-of-the-art performance across major VLN benchmarks, while reducing inference cost and demonstrating promising sim-to-real generalization.

### Acknowledgment

This work was partially supported by the National Natural Science Foundation of China (No. U2441285, No. 62222605, No. U25A20533, No. 62276129, No. 62506165), New Generation Artificial Intelligence-National Science and Technology Major Project (No. 2025ZD0122903), the Natural Science Foundation of Jiangsu Province (No. BK20250082), the Fundamental Research Funds for the Central Universities (No. NE2025010, No. NS2025038), the Jiangsu Funding Program for Excellent Postdoctoral Talent (No. 2025ZB306), and the Natural Science Foundation of Shandong Province (No. ZR2025QC1592).

### Impact Statement

This work presents IDEAL-VLN, a vision-and-language navigation method that decomposes long-horizon instructions into step-wise anchors to improve instruction-following robustness and reduce inference overhead, and introduces an instruction-aligned dataset to support fine-grained instruction–perception alignment. Potential benefits include more reliable indoor assistive and service robotics and accessibility-oriented navigation. Potential harms include dual-use (*e.g.*, facilitating unauthorized navigation or surveillance in private spaces) and physical safety risks if deployed outside validated conditions or under domain shift. We recommend deployment only with system-level safety measures (*e.g.*, collision avoidance, emergency stop, and human oversight) and responsible release practices that discourage misuse.

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

# Appendix

## A. Causal Modeling

In standard VLN settings, given a natural language instruction $\mathcal{L}$ and a history of observations $O_{1:t}$ up to time step $t$, the objective is to learn a policy $\pi_\theta$ that maximizes the likelihood of the action $A_t$. Existing methods typically model the navigation process as a joint conditional probability distribution:

$$A_t \sim P_\theta(A \mid O_{1:t}, \mathcal{L}). \tag{7}$$

In this model, the full instruction $\mathcal{L}$ is treated as a static global condition, exerting a continuous and direct influence on action generation at every time step. From the perspective of a Causal Graph, this modeling implies the existence of a Direct Path from $\mathcal{L}$ to $A_t$. According to Mediation Analysis theory, the Total Effect (TE) of $\mathcal{L}$ on $A_t$ can be decomposed into:

$$TE = \underbrace{NIE(\mathcal{L} \to S_t \to A_t)}_{\text{Desirable Reasoning}} + \underbrace{NDE(\mathcal{L} \to A_t)}_{\text{Harmful Interference}}. \tag{8}$$

Here, NIE (Natural Indirect Effect) represents the Desirable Reasoning path, where the model first extracts the current intent from $\mathcal{L}$ and subsequently guides the action based on this intent. Conversely, NDE (Natural Direct Effect) represents the scenario where the model bypasses explicit reasoning, directly matching lexical features from non-current segments of $\mathcal{L}$ with current visual features. Since $\mathcal{L}$ contains text describing both past and future trajectories, when an agent is located in a "study," VLM models tend to exploit NDE to capture spurious correlations between high-frequency words like "bedroom" and the visual scene. Consequently, $\mathcal{L}$ degenerates into a confounder, introducing significant semantic noise that leads to Hallucinations or the skipping problem during long-horizon navigation.

Our proposed IDEAL-VLN paradigm reformulates the navigation task as a causal inference chain incorporating an explicit Information Bottleneck, implementing intervention via Back-door Adjustment and the do-operator. As demonstrated in Section 3.1, during Semantic Anchoring, $S_t$ is optimized as a Sufficient Statistic of the full information. This maximizes the mutual information $I(S_t; A_t)$ with the current action $A_t$ while minimizing mutual information with non-current instruction segments. Consequently, we enforce a Conditional Independence Constraint during the action alignment phase by implementing causal intervention:

$$A_t \perp \mathcal{L} \mid S_t, I_t. \tag{9}$$

Through this intervention, the influence of the global instruction $\mathcal{L}$ on the action $A_t$ is constrained to be transmitted exclusively through the mediator variable $S_t$. This not only eliminates non-causal spurious correlations but also forces the model to focus entirely on the geometric alignment between the "current vision $I_t$" and the "current intent $S_t$," thereby achieving immunity to instruction interference.

Based on the aforementioned causal model, we define the training objective of IDEAL-VLN as maximizing the Joint Log-Likelihood of this causal chain. Leveraging a single set of MLLM weights $\theta$, we simultaneously optimize the two sub-distributions of Semantic Anchoring and Action Alignment:

$$\mathcal{J}(\theta) = \mathbb{E}_{\mathcal{D}} \left[ \underbrace{\log P_\theta(S_t \mid O_{1:t}, \mathcal{L}, \rho_{decomp})}_{\text{Phase 1: Mediator Inference}} + \lambda \cdot \underbrace{\log P_\theta(A_t \mid I_t, S_t, \rho_{act})}_{\text{Phase 2: Intervened Execution}} \right]. \tag{10}$$

Here, the first term ensures the semantic accuracy of the mediator $S_t$, while the second term guarantees the robustness of action execution after blocking interference. This end-to-end modeling based on the causal graph allows IDEAL-VLN to intrinsically possess anti-interference and explicit reasoning capabilities without introducing external planning modules.

## B. Context Length Paradox

To further substantiate the existence of instruction interference and spurious correlations, we conduct an in-depth analysis of the sensitivity of existing end-to-end paradigms to the historical visual context length $N_{ctx}$. In standard sequence modeling, intuition suggests that longer historical information should facilitate state estimation. However, existing research and our experimental observations reveal a counter-intuitive context length paradox.

In the R2R dataset (Anderson et al., 2018), characterized by shorter paths, the performance of end-to-end models peaks at $N_{ctx} = 8$ but significantly deteriorates when increased to 10. Conversely, in the RxR dataset (Ku et al., 2020), which

features longer paths, the results at $N_{ctx} = 10$ are marginally superior to those at $N_{ctx} = 8$. This phenomenon exposes the pathological mechanism of the NDE, i.e., $NDE(\mathcal{L} \to A_t)$, within end-to-end models.

In the formulation $A_t \sim P_\theta(A \mid I_{t-N:t}, \mathcal{L})$, historical frames $I_{t-N}$ should theoretically serve only to aid localization. However, due to the lack of explicit causal blocking, redundant frames develop spurious correlations with non-current segments of the global instruction $\mathcal{L}$. For instance, a "table" observed at time $t$ might coincide with a "table" described in a future instruction. This spatiotemporal misalignment in feature matching is driven by statistical co-occurrence rather than logical reasoning. Consequently, when $N_{ctx}$ exceeds the physical range required for current local decisions, excessive historical noise becomes a confounder, increasing the variance of action prediction. Models are forced into a difficult trade-off between "long memory for planning" and "short memory for control", making it challenging to accommodate navigation tasks of varying lengths simultaneously.

We decompose the historical visual observation $H_t = I_{t-k:t}$ into two subsets: the relevant context $H_{rel}$ (recent frames for local collision avoidance and geometric alignment) and the nuisance context $H_{nuis}$ (distant historical frames). In end-to-end training, models tend to maximize the mutual information between $A_t$ and the global instruction $\mathcal{L}$. However, $\mathcal{L}$ often contains descriptions of objects relevant to non-current moments. As $k$ increases, redundant context $H_{nuis}$ is introduced. If $H_{nuis}$ happens to contain objects similar to future descriptions in $\mathcal{L}$, the attention mechanism establishes erroneous connections. We model this interference as confounding bias:

$$\text{Bias}(A_t) \propto \sum_{w \in \mathcal{L}_{future}} \sum_{v \in H_{nuis}} \text{Sim}(w, v). \tag{11}$$

In the R2R dataset, where scene repetition is low, the localization benefit provided by $H_{nuis}$ is outweighed by the confounding bias it introduces. Therefore, a shorter $k$ physically shields against interference, yielding better performance. Conversely, in the RxR dataset, where agents are prone to getting lost, reliance on a longer $H_{nuis}$ for global State Estimation is mandatory. Thus, end-to-end models cannot simultaneously satisfy the contradictory requirements of long history for localization and short history for control within a single joint distribution.

To address these issues, IDEAL-VLN resolves the context length paradox through the structural decoupling of the causal chain. In the Semantic Anchoring phase, the confounding factor is transformed into a beneficial evidence variable. The model utilizes it to confirm passage through previous waypoints, thereby inferring the sub-instruction $S_t$ to be executed currently. In the Action Alignment phase, we utilize $do(S_t)$ to block global instruction interference and strictly limit visual input to a short history window. This design renders the action generation distribution $P(A)$ invariant to the total path length. Whether in short-haul R2R or long-haul RxR scenarios, low-level control consistently focuses only on what is currently seen and what needs to be done immediately. Through this mechanism, IDEAL-VLN eliminates spurious correlations arising from redundant frames, achieving true length generalization without the need to manually tune memory length for specific datasets.

## C. Additional Implementation Details

### C.1. Instruction-Aligned Navigation Dataset

To obtain high-quality decomposed sub-instructions $S_t$ for each image, we design a hybrid annotation pipeline. First, we use Qwen3-VL-235B (Bai et al., 2025a) model as an automated annotator. We input the complete navigation instruction $\mathcal{L}$ and the image into the model, prompting it to extract the sub-segment specifically describing the current visual scene and action intent from the full instruction. For RxR data sources (Ku et al., 2020), we further use the timestamp information from the original annotations for temporal consistency verification. Finally, human annotators screen to identify skipped steps and output anomalies. The resulting dataset not only provides precise $I_t \leftrightarrow S_t$ mappings but also retains full context information, offering a solid data foundation for training VLN models with explicit instruction-understanding capabilities.

### C.2. More Details of Model Training and Inference

The training framework consists of two phases: basic action learning with instruction alignment, and error-correction fine-tuning. The primary objective of the first phase is to endow the model with instruction alignment capabilities and establish fundamental cross-modal alignment with expert trajectories. The expert trajectory dataset is constructed using the Habitat simulator, derived from the R2R-CE and RxR-CE datasets. During data collection, to prevent the model from overfitting to specific textural features and to enhance robustness against sensor noise, we implement online photometric

*Table 7.* Image augmentation strategies and their corresponding parameters used in the SFT stage.

| Augmentation Strategy | Parameter |
|---|---|
| Random Brightness | factor = 0.2, prob = 0.5 |
| Random Saturation | factor = 0.2, prob = 0.5 |
| Posterization | bits = 4, prob = 0.5 |
| Random Sharpness | factor = 0.2, prob = 0.5 |
| AutoContrast | prob = 0.3 |
| Gaussian Blur | prob = 0.2 |
| Random Masking | prob = 0.2 |

and geometric augmentations on first-person RGB images during training. As detailed in the Table 7, these strategies include Gaussian blur to simulate motion blur, random masking to mimic occlusions, and color jittering. Furthermore, we employ a "reverse trajectory augmentation" strategy to expand and enrich the navigation paths. Specifically, we utilize Qwen3-235B(Bai et al., 2025a) to semantically rephrase the original instructions and generate corresponding reverse navigation instructions (swapping the start and end points), thereby increasing path diversity.

In the error-correction fine-tuning phase, we employ the SFT-trained model, denoted as $\pi_{SFT}$, to perform autonomous exploration within the simulator environment. When the model deviates from the reference trajectory, we introduce the Shortest Path Follower (SPF) as an expert policy to provide optimal action labels that guide the model back to the target. This mechanism enables the model to learn self-correction in out-of-distribution states, effectively closing the robustness loop of the causal reasoning chain. For the ScaleVLN dataset (Wang et al., 2023b), we adopt the traditional full-instruction input method for training. This mixed training strategy not only preserves the model's ability to process long-context text but also facilitates the learning of Semantic Context Retrieval in hard cases, serving as an effective supplement to the decomposition mechanism.

The inference process of IDEAL-VLN strictly adheres to a "think before acting" causal chain, with specific optimizations for computational efficiency and temporal coherence. During the Thinking Phase, we uniformly sample 8 frames from the historical observation sequence $O_{1:t}$ to serve as visual context input. Conversely, in the Action Phase, we discard long-term memory dependencies and utilize only the most recent 3 frames as local visual references, significantly reducing GPU memory overhead. During the thinking phase, the model may erroneously interpret distant visual cues at the center of the field of view as the current task target, leading to step skipping. To address this, we introduce a coherence constraint on adjacent instructions during inference; specifically, we verify whether the currently generated sub-instruction and the previously generated sub-instruction are coherent within the global instruction context. If an incoherence is detected, the missing instructions are automatically completed to ensure continuity.

We employ five widely established metrics to comprehensively evaluate navigation performance, covering task success, efficiency, and path fidelity: (1) Success Rate (SR): The percentage of episodes where the agent stops within 3 meters of the target. (2) Success weighted by Path Length (SPL): A composite metric that balances success rate with path efficiency, penalizing successful but circuitous paths. (3) Navigation Error (NE): The average geodesic distance between the agent's final stopping location and the goal. (4) Oracle Success (OS): The success rate calculated as if the agent had stopped at the point on its trajectory closest to the goal. (5) Normalized Dynamic Time Warping (nDTW): A measure of the fidelity between the predicted path and the expert trajectory, capturing spatiotemporal similarity.

## D. More Details of Real-World Deployment

To validate the effectiveness of our approach, we construct a mobile manipulation platform based on a quadruped robot. We use the Unitree Go2 as the underlying mobile chassis. For perception, we adopt an ego-centric vision scheme, mounting a RealSense D455 camera on the front of the robot to capture RGB images. Considering the substantial VRAM and computational resources required for large-model inference, we design an efficient edge-cloud collaborative architecture to balance inference accuracy with system response speed. On the robot's edge side, a Jetson Orin module executes a lightweight Automatic Speech Recognition (ASR) model. Upon receiving voice commands from a human operator, the system transcribes them into text instructions and uploads them, along with the current ego-centric RGB image $I_t$, to the server. The server is equipped with a single NVIDIA RTX 4090 GPU. This server hosts our IDEAL-VLN model and transmits the generated action sequences back to the robot for execution; the entire response loop is completed within 0.2 seconds. To rigorously evaluate the generalization boundaries of the model, our navigation policy $\pi_\theta$ is trained entirely on

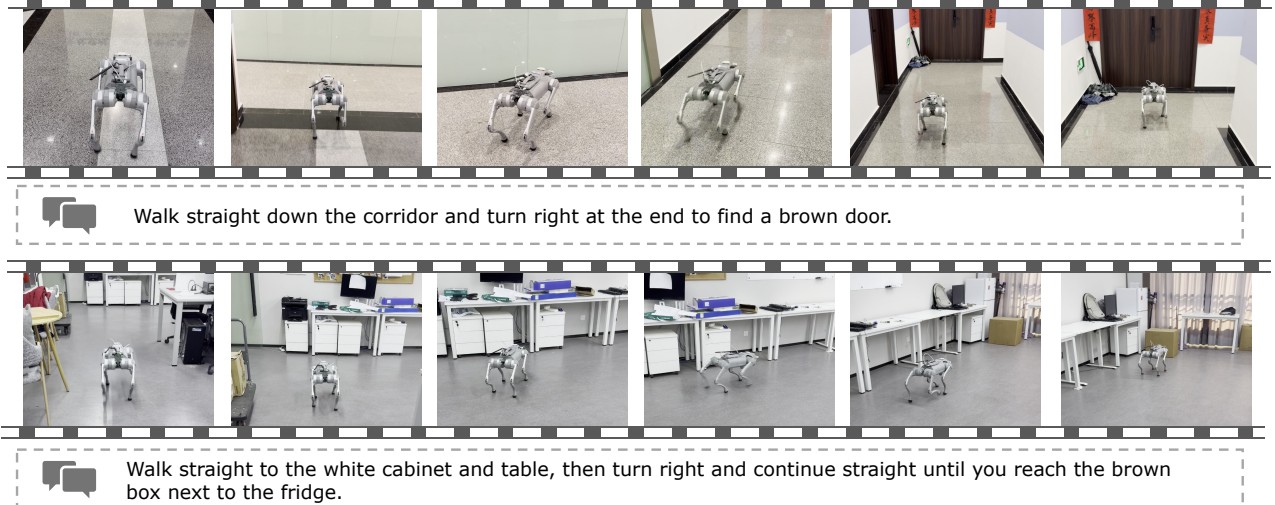

Walk straight down the corridor and turn right at the end to find a brown door.

Walk straight to the white cabinet and table, then turn right and continue straight until you reach the brown box next to the fridge.

*Figure 6.* More real-world deployment results of IDEAL-VLN. The robot successfully follows natural language instructions in unseen indoor environments, grounding visual landmarks and executing long-horizon navigation behaviors under physical constraints.

synthetic data within the Habitat simulator. Consequently, there exists a significant distribution shift between the training set $\mathcal{D}_{sim}$ and the real-world test scenarios $\mathcal{D}_{real}$ in terms of visual appearance and physical properties. This setup aims to verify whether the model has learned domain-invariant navigation representations through large-scale training in the simulated environment. We present the real-world results in Figure 6. Our experiments demonstrate that, despite being trained solely in a simulation environment, IDEAL-VLN exhibits remarkable photometric invariance. The augmentation strategies employed during training enable the model to adapt to photometric noise and rely on geometric structures and semantic landmarks for decision-making.

## E. Limitations and Future Work

Despite its effectiveness, IDEAL-VLN still has several limitations. ***First***, our current system relies on a 7B-parameter multimodal large model. Although the "Think-Before-Act" mechanism reduces memory overhead during action generation, real-time inference with a model of this scale remains challenging on resource-constrained embedded platforms. In future work, we plan to explore lightweight navigation models, especially MLLMs with fewer than 2B parameters, to enable fully onboard real-time inference while preserving navigation accuracy. ***Second***, real-world natural language instructions, particularly those in the RxR, often contain significant amounts of unstructured noise, such as ambiguous references, hesitations, and self-talk. Although IDEAL-VLN alleviates this issue through instruction decomposition, purely textual denoising remains challenging. Our preliminary experiments with an explicit Text Denoising Module reveal a challenging trade-off: aggressive denoising can remove distracting content, but may also discard critical landmark descriptions or topological constraints, leading to semantic discontinuity in the sub-instruction sequence and degraded navigation success rate. In future work, we plan to incorporate a visual-cue augmented reasoning mechanism into the thinking phase. Specifically, when the model encounters ambiguous instructions, it will not rely solely on textual context but will actively integrate the surrounding environmental context to perform multimodal disambiguation. For instance, by identifying unique salient objects within the environment to align with ambiguous directional descriptions, the model can resolve instructional ambiguities without sacrificing critical information.

