# OpenReview forum: "Instruction Decomposition and Action Alignment for Vision-Language Navigation"
_ICML.cc/2026/Conference — ICML 2026 regular_

### Official Review · Reviewer_6k8X · 2026-03-07

**Soundness:** 2
**Presentation:** 2
**Significance:** 1
**Originality:** 1
**Overall Recommendation:** 3
**Confidence:** 4

**Summary:**

This paper proposes IDEAL-VLN (Instruction DEcomposition and Action aLignment), a Vision-and-Language Navigation framework that reframes VLN as a two-step “causal inference chain”: (i) Semantic Anchoring, where the model decomposes the global instruction into an immediate sub-instruction (semantic anchor), and (ii) Action Alignment, where actions are generated conditioned only on the current observation and that anchor. To make the “think-before-act” design practical, the paper introduces an entropy/action-pattern based trigger for when to refresh the sub-instruction, and a Hierarchical Correction Mechanism (HCM) that combines semantic-level thought correction (via a word-overlap criterion) with a spatially-aware intervention probability based on geodesic distance. Experiments on R2R-CE and RxR-CE Val-Unseen suggest strong performance, with reported SR up to 59.8% (R2R-CE) and 59.1% (RxR-CE).

**Compliance With Llm Reviewing Policy:**

Affirmed.

**Final Justification:**

My main remaining concern is still the level of novelty which has not been fully verified. The rebuttal is helpful and resolves the concerns only partially, so I select “partially resolved” and raise my score to weak reject.

**Key Questions For Authors:**

Please explain or clarify the aforementioned weakness and highlight the technical innovation of this paper.

**Limitations:**

Yes

**Strengths And Weaknesses:**

## Strength
1. The design explicitly separates “instruction understanding” (anchor inference) from “execution,” and implements an architectural/prompt-level blocking of full-instruction conditioning during action generation, which is a plausible mechanism to reduce cross-temporal instruction noise.
2. The paper provides ablations on trigger mechanisms (LLM-confidence, SigLIP-based, entropy/action-pattern based) and on correction strategies (DAgger vs STC vs HCM), which helps isolate components and lends credibility to the claimed contributions.
3. Instruction interference under long contexts and the compute burden of visual token history are practical pain points, and the proposed decomposition/bottleneck is relevant to both research and potential robotics applications.

## Weakness
1. The semantic-level thought correction uses a word-overlap ratio with threshold τ (e.g., τ=0.8) to decide whether the predicted sub-instruction is “correct.” This could mis-score paraphrases/synonyms and may encourage unnatural anchors optimized for lexical overlap rather than true intent alignment.
2. The method relies on decomposition/execution prompts (ρ_decomp, ρ_act), yet the exact prompt templates (and decoding settings) are not included in the main text.
3. The paper asserts reduced inference cost by limiting history (e.g., 8 frames in thinking vs 3 frames in acting), but does not present systematic runtime, token count, or GPU memory comparisons against baselines under matched settings.
4. The robot deployment description is interesting (edge-cloud loop, sensors, sim-only training), but the evaluation appears to rely on qualitative figures rather than reporting success rates or failure modes.
5. The overall architecture lacks technical innovation and instead resembles an application of existing standard MLLM frameworks. Additionally, the presentation in Figure 1 should be optimized.

---

> ### Author Rebuttal · Authors · 2026-03-30
>
> > **Q1: Word-overlap ratio with threshold (e.g., τ=0.8) could mis-score paraphrases/synonyms and may encourage unatural anchors.**
>
> **R**: Thank you for the thoughtful comment. We agree word-overlap ratio (WOR) metric can be sensitive to paraphrases. However, our $\tau$ threshold acts as a lightweight safety gate rather than a full semantic measure. Navigation instructions rely on structured directional vocabulary and specific nouns, naturally limiting paraphrasing. As shown below, Sentence-BERT yields negligible performance gains (+0.2% SR) despite significant computational overhead, proving lexical overlap efficiently captures the necessary signal. We will clarify this choice and add these results to the revision.
> |Method|Extra Mem.| Extra FLOPs| SR|
> |-|-|-|-|
> |Sentence-BERT| ~500MB| ~10G | 56.1%|
> |WOR($\tau=0.7$)|  None | None | 55.8%|
> |WOR($\tau=0.8$)|  None | None | 55.9%|
> |WOR($\tau=0.9$)|  None | None | 55.2%|
> > **Q2: The exact prompt templates and decoding settings.**
>
> **R**: We thank the reviewer for the constructive feedback.
>
> - **Decomposition prompt ($\rho_{decomp}$} :**
>
> ```
> You are an autonomous navigation assistant. The complete instruction is <instruction>. These are your historical observations: <history>. Please respond to the subcommand corresponding to the current observation <image>.
> ```
> - **Action prompt ($\rho_{act}$} :**
>
> ```
> Your task is to <sub_task>. Based on the current observation <image>, devise an action sequence to follow the instruction using the four actions: Turn Left, Turn Right, Forward and Stop.
> ```
>
> We would like to further clarify that IDEAL-VLN is  robust to prompt templates, yielding negligible performance differences across variants.  For decoding setting, we use the same generation settings as the baselines, retaining the default configuration from the `transformers` library.
> We will include these details in the revision.
>
> > **Q3: Lack systematic baseline comparisons for runtime, token counts, and GPU memory.**
>
> **R:** Quantitatively, IDEAL-VLN cuts costs by dropping full-context memory during action alignment (requiring only ~2K tokens/step). In an average 21-step R2R episode, it triggers full-context thinking ~5 times. This halves token usage and prefill FLOPs compared to JanusVLN, while improving SR. Furthermore, on an RTX 3090, action inference is 4× faster, [with thinking overhead masked by agent motion](https://anonymous.4open.science/r/IDEAL-VLN/think.jpg).
>
> |Stage| Input Tokens/Step| Prefill Runtime(ms)/Step| Steps/Episode| Total Tokens/Episode | Prefill FLOPs/Episode| SR|
> |-|-|-|-|-|-|-|
> |JanusVLN (action)| 7k | ~1625| 21 |  147K| ~2310T| 42.7% |
> |IDEAL-VLN (think)| 7k | ~1832| 5 | 35K| ~550T | - |
> |IDEAL-VLN (action)| 2k | ~462 |21 | 42K| ~630T| 51.1%|
> >   **Q4: Quantitative success rates and failure modes of the robot deployment evaluation.**
>
> **R:** Thank you for the valuable suggestion. We have conducted quantitative evaluations. In 30 real-world trials on unseen indoor and outdoor scenes as shown in table, IDEAL-VLN achieves **86.7% SR indoors** and **76.7% SR outdoors**. Indoor failures arise from dynamic obstacles and glass reflections, while outdoor failures are largely caused by lighting variation. These results show promising sim-to-real transfer but also reveal gaps in illumination and material effects. We will include them in the revision.
> |Scene| Success | Failure | SR |
> |-|-|-|-|
> |Indoor| 26 | 4 | 86.7% |
> |Outdoor| 23 | 7 | 76.7% |
> > **Q5: The architecture lacks technical novelty, resembling standard MLLM applications. Additionally, Figure 1 requires optimization.**
>
> **R:** We thank the reviewer for the constructive comments. We respectfully clarify that IDEAL-VLN structurally reformulates VLN beyond standard MLLM applications:
> - **Problem Modeling:** Unlike traditional end-to-end policies ($A_t \sim P_\theta(A\mid O_{1:t},L)$) , we reformulate VLN as a two-stage causal chain (semantic anchoring $\rightarrow$ action alignment). By explicitly blocking the direct path $L \to A_t$ , the model avoids spurious correlations from non-current instruction segments. This strictly decouples long-range language understanding from local action control.
> -  **Training Mechanism:** Departing from standard SFT, our Hierarchical Correction Mechanism (HCM) jointly corrects intermediate thoughts and actions. It combines semantic-level thought correction with spatially-aware adaptive intervention, which is essential for stabilizing the "Think-Before-Act" paradigm.
> - **Efficiency Improvement:**  As shown in the response to Q3, token usage drops by ~50% while improving performance.
> - **Data Contribution:** We  introduce  a large-scale dataset designed to support fine-grained semantic-action alignment, which will  be open-sourced along with code.
>
> As suggested, we will optimize Figure 1 to more clearly illustrate the causal inference chain and the unified "Think-Before-Act" paradigm. We appreciate the reviewer’s helpful feedback.

---

> > ### Author Rebuttal · Reviewer_6k8X · 2026-04-03
> >
> > My main remaining concern is still the level of novelty which has not been fully verified. The rebuttal is helpful and resolves the concerns only partially, so I select “partially resolved” and raise my score to weak reject.

---

> > > ### Author Response · Authors · 2026-04-03
> > >
> > > Thank you for your follow-up and for the thoughtful consideration. We are pleased that our responses have partially addressed your concerns, and **we sincerely appreciate your reassessment of the score**.
> > >
> > > Regarding the remaining concern on novelty, we would like to further clarify that the contribution lies in structurally reformulating the VLN task into a novel framework, rather than an application of existing MLLM techniques. Specifically, it introduces a new decision-making paradigm, training mechanism, and  dataset contribution. We provide a more detailed clarification below:
> > >
> > > - **Problem Modeling**: We introduce **Structural Causal Models** into end-to-end VLN tasks, explicitly decoupling the navigation process into two distinct phases: "semantic anchoring" and "action alignment",  to the best of our knowledge,  which **have not been introduced in prior work**. During the action generation phase, we implement Causal Intervention by executing the $do(S_t)$ operator, which structurally blocks the direct feedback path from the global instruction $\mathcal{L}$ to the action $A_t$. This ensures that the model strictly adheres to the conditional independence constraint $A_t \perp \mathcal{L} \mid S_t$, thereby eliminating the semantic noise and interference caused by long-sequence instructions at the causal root. This fundamentally resolves the issues of spurious correlations and semantic hallucinations prevalent in existing end-to-end MLLM navigation models, which map actions by treating the complete text instruction as a global static condition ($A_t \sim P_\theta(A \mid O_{1:t}, \mathcal{L})$).
> > > - **Training Mechanism**: We achieve a joint optimization of cognition and behavior.
> > > The proposed  **Think-Before-Act** paradigm that directly blocks the propagation of erroneous thoughts along the causal inference chain. Furthermore, we introduce Spatially-Aware Adaptive Intervention, which models the expert intervention probability as a non-linear function of the current cross-track error. This effectively balances the model's capacity for in-distribution exploration and out-of-distribution safety fallback in unseen environments.
> > > - **Efficiency Improvement**: Through structural decoupling, IDEAL-VLN relies solely on local visual references during the action execution phase, completely discarding the reliance on long-horizon visual history. Experimental results demonstrate that action generation requires only 462ms per step, approximately **one-fourth of the time required by existing methods** that enabling real-time navigation. While traditional MLLM thinking stage introduces additional latency, our instruction-decoupled **thinking latency is effectively masked by the agent's physical movement time**. The agent thinks about the next sub-instruction while moving, and upon reaching the designated location, generates the next action based on the new sub-instruction.
> > > - **Data Contribution**: Existing VLN datasets typically provide only trajectory-level instruction annotations, severely lacking frame-level correspondences. To address this, we construct and will release the large-scale Instruction-Aligned Navigation Dataset, containing 160K image-sub-instruction pairs. This acts as a critical catalyst in driving the field from coarse-grained navigation toward fine-grained semantic-action alignment.
> > >
> > > The ablation studies presented in the main paper demonstrate the effectiveness of these modules, with each component improving overall model performance. When compared to existing VLN models under the same MLLM backbone, our model achieves leading results across various navigation benchmarks. This validates that the performance improvements stem directly from the proposed modeling and training mechanisms.
> > >
> > > |Think-before-Act| HCM| NE | SR | SPL| Prefill Runtime(ms)/Step|
> > > |:-:|:-:|:-:|:-:|:-:|:-:|
> > > | | | 6.83| 42.7| 37.8|  1625|
> > > |$\surd $| | 6.12| 51.1| 45.9|   462|
> > > |$\surd $| $\surd $| 5.68| 55.9|48.2|   462|
> > >
> > >
> > > We hope this clarifies the novelty and contribution of our work. We sincerely appreciate the reviewer’s constructive feedback, which has helped improve our work.

---

### Official Review · Reviewer_Z6EP · 2026-03-12

**Soundness:** 3
**Presentation:** 3
**Significance:** 3
**Originality:** 3
**Overall Recommendation:** 4
**Confidence:** 4

**Summary:**

This paper introduces IDEAL-VLN, a navigation paradigm that reformulates Vision-and-Language Navigation as a causal inference chain to decouple complex instruction understanding from real-time execution. By implementing a "Think-Before-Act" mechanism, the agent proactively decomposes long-horizon instructions into atomic sub-tasks, effectively mitigating semantic interference and navigation hallucinations. To optimize this process, the authors propose a hierarchical correction mechanism that jointly trains the agent's cognitive reasoning and behavioral alignment, ensuring robust performance in complex, unseen environments.

**Compliance With Llm Reviewing Policy:**

Affirmed.

**Final Justification:**

I appreciate the authors' responses and the rebuttal has addressed my concerns adequately.

**Key Questions For Authors:**

Refer to the weaknesses

**Limitations:**

yes

**Strengths And Weaknesses:**

Strengths:
1. By reformulating VLN as a causal inference chain that explicitly separates semantic anchoring from action alignment, the paper provides a mathematically grounded solution to the persistent problem of instruction-induced semantic noise.
2. The framework shifts the paradigm from fragile end-to-end mapping to proactive, modular planning, offering a highly scalable and interpretable architecture for deploying complex MLLMs in real-world embodied navigation.
3. The "Think-Before-Act" mechanism, which treats sub-instructions as latent variables to reduce task dimensionality, represents a creative application of causal reasoning that effectively breaks the traditional, passive navigation bottleneck.
4. The paper is  well-structured, clearly distinguishing between cognitive reasoning and behavioral execution, which makes the proposed adaptive triggering and hierarchical correction mechanisms intuitive and easy to follow.

Weaknesses:
1.  While the "adaptive triggering strategy" claims to be "zero-overhead," the introduction of a reasoning step (thinking) before every action inherently increases the latency of the closed-loop control cycle.  It would be better to provide more details for extra overhead introduced by reasoning token.
2.  The "hierarchical correction mechanism" bears similarities to existing "Chain-of-Thought" (CoT) fine-tuning strategies in LLMs. The authors need to more clear classification.

---

> ### Author Rebuttal · Authors · 2026-03-30
>
> > **Q1: While the "adaptive triggering strategy" claims to be "zero-overhead," the introduction of a reasoning step (thinking) before every action inherently increases the latency of the closed-loop control cycle. It would be better to provide more details for extra overhead introduced by reasoning token.**
>
> **R**：We thank the reviewer for this valuable feedback. We address and clarify this concern from the following  perspectives.
>
> **First**, we clarify that the term "*zero-overhead*"  specifically refers to the  adaptive triggering strategy itself, which does not introduce additional parameters or inference overhead, unlike LLM-based or SigLIP-based approaches.  As shown in Table 3 of the manuscript, while the SigLIP-based method achieves slightly better performance, it incurs extra GPU memory usage (\~0.9G). In contrast, our entropy-based triggering achieves competitive performance without  any additional modules.
>
> |Method| Extra Mem. | NE | SR | SPL|
> |-|-|-|-|-|
> |Vanilla| / | 6.83 | 42.7 | 37.8 |
> |LLM-based| / | 7.05 | 35.8 | 29.2|
> |Siplip-based| 0.9G| 6.18 | **51.5** | **46.0** |
> |Entropy-based (Ours)| / | **6.12** | 51.1| 45.9|
>
> **Second**, regarding the concern about increased latency introduced by the reasoning("thinking") step, we  would like to clarify that IDEAL-VLN does not perform this "thinking" at every step.  Instead, the thinking stage is triggered sparsely (\~5 times per episode), while lightweight action alignment runs at every step.
>
> From an overall efficiency perspective, our IDEAL-VLN is in fact more efficient than full-instruction conditioning. As shown below (R2R subset), under the same setting as JanusVLN,  JanusVLN processes \~7K tokens per step (\~147K tokens per episode), whereas IDEAL-VLN reduces this to\ ~77K tokens (\~48% reduction) by replacing high-frequency global conditioning with low-frequency reasoning. Similarly, the total prefilling FLOPs are reduced from \~2310T to \~1180T (≈50% reduction), while achieving higher success rate (SR). On RTX 3090, our method achieves 4× faster action inference than existing methods, [with thinking overhead masked by agent motion](https://anonymous.4open.science/r/IDEAL-VLN/think.jpg).
>
> |Stage| Input Tokens/Step| Prefill Runtime(ms)/Step| Steps/Episode| Total Tokens/Episode | Prefill FLOPs/Episode| SR|
> |-|-|-|-|-|-|-|
> |JanusVLN (action)| 7k | ~1625| 21 |  147K| ~2310T| 42.7% |
> |IDEAL-VLN (think)| 7k | ~1832| 5 | 35K| ~550T | - |
> |IDEAL-VLN (action)| 2k | ~462 |21 | 42K| ~630T| 51.1%|
>
> **Finally**, in real-world deployment, the reasoning latency is effectively **hidden by actuation time**. The model outputs short action sequences that require \~3–5 seconds for physical execution, while sub-task reasoning (\~2 seconds) is performed concurrently. Therefore, the reasoning step does not increase the closed-loop latency in practice.
>
> Overall, IDEAL-VLN reduces total computation by \~50% while maintaining efficient real-time execution. We will clarify these points in the revision.
>
> > **Q2: The "hierarchical correction mechanism" bears similarities to existing "Chain-of-Thought" (CoT) fine-tuning strategies in LLMs. The authors need to more clear classification.**
>
> **R**: We thank the reviewer for pointing out this connection. Our Hierarchical Correction Mechanism (HCM) is fundamentally different from CoT fine-tuning in the following aspects:
>
> (1)  **Training Paradigm:** HCM performs *online correction* during closed-loop embodied rollouts, whereas CoT fine-tuning primarily relies on sequence supervision over static data. In embodied navigation, the agent must continuously interact with the environment and recover from off-trajectory states. Standard CoT supervision, which relies on offline reasoning traces, is not designed to handle such distribution shifts or closed-loop corrections. In contrast, HCM performs online correction during policy execution, jointly updating both reasoning  (sub-instruction)  and actions.
>
> (2) **Supervision Target:** HCM supervises task-relevant state variables rather than free-form natural language rationales. Specially, the Semantic-level Thought Correction (STC) module corrects the sub-instruction $S_t$ aligned with the agent’s current state, which directly serves as an intermediate control variable for action generation. In contrast, CoT fine-tuning focuses on training the model to generate explanatory reasoning text.
>
> (3) **Components:** HCM consists of STC and Spatial-level Adaptive Intervention (SAI). STC module can be viewed as latent-plan supervision, while SAI dynamically adjusts expert intervention probability based on the geodesic cross-track error, forming a state-dependent, spatially-aware correction policy. This type of adaptive intervention has no direct counterpart in standard CoT training. Empirically, Table 4 in the main paper further demonstrates the effectiveness of each component.

---

> > ### Author Rebuttal · Reviewer_Z6EP · 2026-04-03
> >
> > I appreciate the authors' responses and the rebuttal has addressed my concerns adequately.

---

> > > ### Author Response · Authors · 2026-04-03
> > >
> > > Thank you for your kind feedback. We sincerely appreciate your thoughtful review and are glad that our responses have addressed your concerns. We will incorporate the discussed clarifications into the final version.

---

### Official Review · Reviewer_3auh · 2026-03-13

**Soundness:** 3
**Presentation:** 3
**Significance:** 3
**Originality:** 3
**Overall Recommendation:** 5
**Confidence:** 3

**Summary:**

The paper proposes IDEAL-VLN, which reformulates VLN as a two-stage causal chain - semantic anchoring to infer the current sub-instruction from the instruction and observation history, and action alignment to generate the next action conditioned on the current RGB image and the anchor. This formulation is intended to block irrelevant parts of the instruction from directly affecting action generation. Moreover, IDEAL-VLN introduces Think-Before-Act mechanism with adaptive triggering. The paper also introduces a new Instruction-Aligned Navigation Dataset with image and sub-instruction pairs collected from MP3D R2R and RxR trajectories.

**Compliance With Llm Reviewing Policy:**

Affirmed.

**Final Justification:**

I thank the authors for addressing my concerns. After carefully reading other reviewers' comments and the authors' rebuttals, I have decided to increase my score. I'd encourage the authors to add the additional clarifications, quantified effiiciency benefits and real-world results in the final revision to make the paper strong.

**Key Questions For Authors:**

1) The paper claims large efficiency benefits but does not quantify them enough. Do you have numbers that could demonstrate this?
2) Do you think a learned semantic similarity metric could replace keyword overlap?
3) I'd also ask the authors if they have real quantitative sim-to-real evaluation, not only qualitative examples.

**Limitations:**

yes

**Strengths And Weaknesses:**

Strength:
In VLN, the agent's next action usually depends on only a sub-instruction and not the full instruction. The paper turns that intuition into a concrete architecture. The dataset contribution is also valuable. Fine-grained alignment between egocentric images and local sub-instructions has been missing from most VLN datasets, and a 160K aligned dataset could be useful to the community if it is released and well curated.

Weakness:
I think that the method is closer to a hierarchical subgoal decomposition or latent sub-instruction policy rather than a causal-inference method. The model predicts an intermediate anchor and then act from it, which is useful. But it is not very clear to me how that establishes causal identification.
Also, the paper claims that full instruction conditioning causes hallucinations, but there is little direct analysis on when and how the full instruction model attends to wrong instruction segments.
The paper lacks quantitative real-world benchmark, which makes the sim-to-real transfer more of a demonstration than an evaluation.

---

> ### Author Rebuttal · Authors · 2026-03-30
>
> > **Q1: Closer to hierarchical subgoal decomposition or latent sub-instruction policy rather than causal inference.**
>
> **R**：Thank you for your thoughtful comment. We would like to clarify that the key difference lies a *strict information bottleneck*.
>
> Unlike hierarchical policies that still expose the global instruction $\mathcal{L}$ to the low-level policy allowing spurious correlations, our method enforces a strict information bottleneck. By applying a causal intervention $do(S_t)$ on the semantic anchor $S_t$, we structurally ensure actions $A_t$ are conditionally independent of $\mathcal{L}$. Furthermore, **Appendix B** demonstrates that $\mathcal{L}$ acts as a confounder driving the *context length paradox*. Our explicit structural decoupling mitigates these long-horizon hallucinations, going beyond standard hierarchy.
> > **Q2:  It is not very clear to me how that establish causal identification?**
>
> **R**： We thank the reviewer for this rigorous question.  Strictly speaking, IDEAL-VLN does not perform strict statistical causal identification. Instead, our method should be viewed as an *intervention-equivalent architecture* designed upon the principles of causal mediation analysis.
>
> In VLN, the instruction $\mathcal{L}$ affects action $A_t$ via the desired Natural Indirect Effect (NIE, $\mathcal{L} \rightarrow S_t \rightarrow A_t$) and the Natural Direct Effect (NDE, $\mathcal{L} \rightarrow A_t$). Because $\mathcal{L}$ contains past and future states, the NDE introduces spurious spatio-temporal correlations. By introducing mediator $S_t$ and an information bottleneck, we effectively block this direct path ($\mathcal{L} \rightarrow A_t$).
>
> We will revise this to clarify this point and frame our method more precisely as a mediation-inspired structural intervention.
> > **Q3: Direct analysis on when or how the full instruction model attends to wrong instruction segments.**
>
> **R**：We will clarify this mechanism in the revision through three key analyses:
> - **How it happens (Appendix B):** Attention mechanisms form spurious cross-spatiotemporal connections when redundant visual context ($H_{nuis}$) superficially resembles non-current instructions in $\mathcal{L}$, leading to confounding bias:
> $Bias(A\_t) \propto \sum\_{w \in \mathcal{L}\_{future}} \sum\_{v \in H\_{nuis}} sim(w, v)$.
>
> - **When it happens (Context Length Paradox):** Hallucinations trigger when context length ($N_{ctx}$) exceeds local control requirements. Empirically, R2R performance peaks at $N_{ctx}=8$ but degrades at 10, confirming excess history exacerbates misaligned attention.
> - **Alignment Failure (Sec 4.6):** Existing models frequently fail to isolate the current task and remain highly susceptible to distant, irrelevant cues. This proves full-instruction conditioning systematically induces hallucination.
>
> > **Q4: Quantify the claimed efficiency benefits.**
>
> **R**：Thank you for the constructive suggestion. Quantitatively, IDEAL-VLN cuts costs by dropping full-context memory during action alignment (requiring only ~2K tokens/step). In an average 21-step R2R episode, it triggers full-context thinking ~5 times. This halves token usage and prefill FLOPs compared to JanusVLN, while improving SR. Furthermore, on RTX 3090, action inference is 4× faster, [with thinking overhead masked by agent motion](https://anonymous.4open.science/r/IDEAL-VLN/think.jpg).
>
> |Stage| Input Tokens/Step| Prefill Runtime(ms)/Step| Steps/Episode| Total Tokens/Episode | Prefill FLOPs/Episode| SR|
> |-|-|-|-|-|-|-|
> |JanusVLN (action)|7k|~1625| 21|147K|~2310T|42.7%|
> |IDEAL-VLN (think)|7k|~1832| 5 |35K|~550T|-|
> |IDEAL-VLN (action)| 2k |~462 |21|42K| ~630T|51.1%|
> > **Q5：Do you think a learned semantic similarity metric could replace keyword overlap?**
>
> **R**：We evaluated Sentence-BERT as a learned semantic similarity metric. However, it yields only marginal gain over keyword overlap (56.1% vs. 55.9% SR) while adding **\~500MB memory and \~10G FLOPs** as shown blow. Since navigation instructions are dominated by directional phrases and landmark nouns, lexical overlap already provides a strong signal. We therefore keep word overlap for simplicity and zero extra overhead, and will add this discussion to the revision.
>
> |Method|Extra Mem.| Extra FLOPs|SR|
> |-|-|-|-|
> |Sentence-BERT|~500MB| ~10G | 56.1%|
> |Word-overlap Rate|None |None|55.9%|
> > **Q6: Real quantitative sim-to-real evaluation.**
>
> Thank you for the valuable suggestion. We have conducted quantitative evaluations. In 30 real-world trials on unseen indoor and outdoor scenes as shown in table below, IDEAL-VLN achieves **86.7% SR indoors** and **76.7% SR outdoors**. Indoor failures arise from dynamic obstacles and glass reflections, while outdoor failures are largely caused by lighting variation. These results show promising sim-to-real transfer but also reveal gaps in illumination and material effects. We will include them in the revision.
> |Scene| Success | Failure |SR|
> |-|-|-|-|
> |Indoor|26|4|86.7%|
> |Outdoor|23|7|76.7%|

---

> > ### Author Rebuttal · Reviewer_3auh · 2026-04-01
> >
> > I appreciate the authors' responses and the rebuttal has addressed my concerns adequately.

---

> > > ### Author Response · Authors · 2026-04-02
> > >
> > > Thank you for your kind feedback. We sincerely appreciate your careful review and are glad that our responses have addressed your concerns. We will incorporate the discussed clarifications into the final version.

---

### Decision · Program_Chairs · 2026-04-30

**Decision:**

Accept (regular)

**Comment:**

IDEAL-VLN reframes navigation as a two-stage causal chain of semantic anchoring and action alignment, utilizing an adaptive "think-before-act" mechanism. Although the paper received mixed reviews, the authors successfully addressed the majority of the concerns raised. While a concern regarding novelty remains, the reviewers did not specify which aspects were considered non-novel or provide references to closely related prior work. I find that the authors provided sufficient clarification across all key dimensions of novelty, including problem modeling, training mechanisms, efficiency improvements, and data contributions.